# Recent Developments in (K, Na)NbO_3_-Based Lead-Free Piezoceramics

**DOI:** 10.3390/mi15030325

**Published:** 2024-02-26

**Authors:** Geun-Soo Lee, Jung-Soo Kim, Seung-Hyun Kim, San Kwak, Bumjoo Kim, In-Su Kim, Sahn Nahm

**Affiliations:** Department of Materials Science and Engineering, Korea University, 145 Anam-ro, Seongbuk-gu, Seoul 02841, Republic of Korea; coolr17@korea.ac.kr (G.-S.L.); khsky1357@korea.ac.kr (J.-S.K.); 94seankim@korea.ac.kr (S.-H.K.); mtsan1225@korea.ac.kr (S.K.); pre4827@korea.ac.kr (B.K.); yskim9552@korea.ac.kr (I.-S.K.)

**Keywords:** KNN-based piezoceramic, lead-free, polymorphic phase boundary, reactive template grain growth, phase transition

## Abstract

(K_0.5_Na_0.5_)NbO_3_ (KNN)-based ceramics have been extensively investigated as replacements for Pb(Zr, Ti)O_3_-based ceramics. KNN-based ceramics exhibit an orthorhombic structure at room temperature and a rhombohedral–orthorhombic (R–O) phase transition temperature (*T_R–O_*), orthorhombic–tetragonal (O–T) phase transition temperature (*T_O–T_*), and Curie temperature of −110, 190, and 420 °C, respectively. Forming KNN-based ceramics with a multistructure that can assist in domain rotation is one technique for enhancing their piezoelectric properties. This review investigates and introduces KNN-based ceramics with various multistructures. A reactive-templated grain growth method that aligns the grains of piezoceramics in a specific orientation is another approach for improving the piezoelectric properties of KNN-modified ceramics. The piezoelectric properties of the [001]-textured KNN-based ceramics are improved because their microstructures are similar to those of the [001]-oriented single crystals. The improvement in the piezoelectric properties after [001] texturing is largely influenced by the crystal structure of the textured ceramics. In this review, [001]-textured KNN-based ceramics with different crystal structures are investigated and systematically summarized.

## 1. Introduction

Piezoelectric materials can convert mechanical energy into electrical energy or vice versa and have been applied to diverse electronic devices, including piezoelectric actuators, ultrasonic transducers, and piezoelectric energy harvesters [1,2,3,4,5,6,7,8,9,10,11,12,13,14,15,16,17,18,19,20,21]. Until recently, Pb(Zr, Ti)O_3_ (PZT)-modified ceramics were widely utilized to fabricate piezoelectric components because of their excellent piezoelectric properties and high Curie temperatures (*T*_C_), allowing them to control the piezoceramics markets [22,23,24,25,26,27,28,29]. However, PZT-modified piezoceramics contain more than 60 wt% PbO, which causes ecological issues. Therefore, the investigations on lead-free piezoceramics commenced at the beginning of this century, seeking to substitute the PZT-modified piezoceramics [30,31,32,33,34,35,36,37]. To date, various lead-free piezoceramics, such as BaTiO_3_ (BT), bismuth-layered structures, and Bi_1/2_Na_1/2_TiO_3_ (BNT), have been studied [38,39,40,41,42,43,44,45,46,47,48,49,50,51,52,53,54,55,56,57]. BaTiO_3_-modified ceramics have a high piezoelectric charge constant (*d*_33_); however, they have a low *T_C_* at ~130 °C, which needs to be enhanced for practical applications. Bismuth-layered structure-based ceramics have a large mechanical quality factor (*Q_m_*) with a high *T_C_*; however, their low electromechanical coupling factor (*k_p_*) and difficulties in densification limit their practical applications [41,42,43,44,45,46]. BNT-modified ceramics have been investigated for applications in actuators owing to their large electric-field-induced strain [47,48,49,50,51,52,53,54,55,56,57]; however, this strain is reduced at high temperatures, which needs to be solved for applications in piezoelectric actuators [58,59,60].

Lead-free (K_1−x_Na_x_)NbO_3_ (KNN) piezoceramics were invented in the mid-1950s [61,62,63]; however, they were not studied extensively because their piezoelectric properties were lower than those of PZT-modified piezoceramics (*d*_33_ = 120 pC/N, *ε^T^*_33_/*ε*_0_ = 500, *k_p_* = 0.35, and *Q_m_* = 500, where *ε^T^*_33_/*ε*_0_ represents the dielectric constant) [64,65,66,67,68]. KNN ceramic has an orthorhombic (O) structure at room temperature (RT) and a rhombohedral–orthorhombic (R–O) phase transition temperature (*T_R–O_*), an orthorhombic–tetragonal (O–T) phase transition temperature (*T_O–T_*), and *T_C_* of −110, 190, and 420 °C, respectively [69,70,71,72,73,74,75,76]. Moreover, a pseudocubic (PC) structure was observed at temperatures near *T_C_* [77,78,79]. The PC structure cannot be considered as a cubic (C) structure because a KNN-modified ceramic with a pure PC structure exhibits piezoelectric properties [80]. Therefore, a PC structure was interpreted as an R structure or the T–C multistructure based on Rietveld analysis [77,78,79,80]. In addition, the PC structure has been understood as various structures depending on the specimens in which the PC structure is developed. Therefore, it is difficult to suggest the unique atomic model for the PC structure developed in the KNN-modified piezoceramics. However, since detailed information on the PC structure was reported in previous studies [77,78,79,80,81,82], the various atomic model diagrams for the PC structure can be obtained using the information reported in the previous studies. KNN-modified lead-free piezoceramics started gaining interest at the beginning of this century and increased considerably after Satio et al. reported that KNN-modified ceramics fabricated by the reactive template grain growth (RTGG) technique provided a high *d*_33_ of 416 pC/N [83]. However, the piezoelectric characteristics of the KNN-modified piezoceramics need further improvements to replace PZT-modified piezoceramics. 

Two methods have been employed to enhance the piezoelectric characteristics of KNN-modified piezoceramics. The first method involves developing KNN-based ceramics with a polymorphic phase transition (PPT) structure wherein multiple structures can coexist [84,85,86,87,88,89,90,91,92,93,94,95,96,97,98,99,100,101,102,103]. The existence of the multistructure flattens the free energy versus polarization curve owing to the presence of many spontaneous polarization directions. Therefore, the domain rotation is easy in the piezoceramics with the multistructure, resulting in the large piezoelectric properties [104,105,106]. A general method to develop the KNN-based piezoceramics with multistructure is to shift the *T_R–O_* and *T_O–T_* to RT with the addition of various elements and solid solutions. Further, nanodomains develop in KNN-based ceramics with a PPT structure, and the domain boundary energy decreases with a decreasing domain size [107,108,109]. Piezoelectric properties were enhanced in KNN-modified ceramics with a PPT structure because of the decreased domain size, which is referred to as an extrinsic effect [110]. The second method is the RTGG method, which lines up the grains of piezoelectric ceramics in a particular direction to improve the piezoelectricity of KNN-related piezoceramics [111]. NaNbO_3_ (NN) seeds are regularly utilized to texture KNN-related piezoelectric ceramics toward the [001] direction [81,112]. A traditional tape-casting method has been employed to fabricate the textured KNN-based piezoceramics. The piezoelectric properties of the [001]-textured KNN-related piezoceramics are improved because their microstructures are alike those of [001]-oriented single crystals [113,114,115]. The enhancement in piezoelectricity after [001] texturing is largely affected by the crystal structure of the ceramics, and therefore, the relationship between crystal structure and piezoelectricity enhancement has also been investigated [116,117,118,119]. 

KNN-modified piezoceramics with various PPT structures have been developed in which two structures coexist: O–T, R–O, R–T, O–PC, and T–PC multistructures. The O–T PPT structure can be formed by decreasing the *T_O–T_* to RT, and the KNN-modified ceramic with the O–T multistructure provided *d*_33_ values ranging from 220–502 pC/N [69,70,79]. KNN-related piezoceramics with an R–O multistructure, which can be developed by increasing the *T_R–O_* to temperatures near RT, showed a relatively low *d*_33_ (~230 pC/N) [71]. Numerous studies have been conducted to fabricate KNN-related piezoceramics with an R–T multistructure by decreasing the *T_O–T_* and increasing the *T_R–O_* to RT at the same time; they showed relatively large *d*_33_ values of 430–550 pC/N [72,74,77]. Finally, KNN-based ceramics with an O–PC or T–PC multistructure were developed by reducing *T_C_* near RT, providing a slightly improved *d*_33_ value of ~330 pC/N. The composition and piezoelectric properties of KNN-modified piezoceramics, in which the two structures coexist, were reviewed in a previous study [69,70,71,72,73,74,75,76,77,78,79,80]. 

Recently, KNN-based piezoceramics containing three structures, such as O–T–PC and R–O–T multistructures, have been reported, and their piezoelectric characteristics are significantly better than those of KNN-modified piezoceramics containing two phases [77,78,82,118,120,121,122]. However, their composition and piezoelectric characteristics are yet to be reviewed. Furthermore, [001]-textured KNN-modified piezoceramics that use NN templates exhibit an excellent *d*_33_ value of 805 pC/N that is larger than or similar to that of PZT-based piezoceramics [118]. However, [001]-textured KNN-modified ceramics are yet to be investigated. 

Therefore, this review summarizes various KNN-modified piezoceramics wherein three structures coexist and discusses their piezoelectric properties. Furthermore, [001]-textured KNN-modified piezoceramics are introduced, and the link between the crystal structure and enhancement of piezoelectric characteristics after [001]-texturing is discussed.

## 2. KNN-Modified Ceramics with the O–T–PC Multistructure

### 2.1. (Na, K)(Nb, Sb)-CaTiO_3_ Ceramics with the O–T–PC (or O–T–C) Multistructure

CuO-added 0.96(Na_0.5_K_0.5_)(Nb_1−x_Sb_x_)O_3_-0.04CaTiO_3_ [NK(N_1−x_S_x_)-CT] piezoceramics with 0.03 ≤ x ≤ 0.1 were well-sintered at 970 °C in the air. For a detailed analysis of the crystal structures of these ceramics, X-ray diffraction (XRD) peaks at ~67° are obtained by slow-speed scanning and deconvoluted using the Voigt function, as shown in Figure 1a–f. Piezoceramics with x = 0.03 exhibit an O–T multistructure (Figure 1a), which changes to an O–T–C structure for the piezoceramic with x = 0.04. An identical structure is observed for the specimen with x = 0.05 (Figure 1b,c). The O structure is removed when x exceeds 0.05, and the piezoceramics with 0.07 ≤ x ≤ 0.1 have a *T*–*C* multistructure (Figure 1d–f). Therefore, the crystal structure of the NK(N_1−x_S_x_)-CT piezoceramic transformed from an O–T multistructure to the O–T–C and T–C multistructures with an increase in the Sb^5+^ content. The PC structure developed in the NK(N_1−x_S_x_)-CT piezoceramics with x > 0.03 was understood as a T–C multistructure consisting of the *P4mm* T and *Pm3m* C structures based on the Rietveld analysis [78]. The diverse electrical characteristics of the NK(N_1−x_S_x_)-CT piezoceramics sintered at 970 °C were investigated, as shown in Figure 2. Piezoceramics (0.04 ≤ x ≤ 0.05), which have the O–T–C multistructure, showed relatively small *d*_33_ values (256–275 pC/N) and *k_p_* values (0.46–0.52); therefore, NK(N_1−x_S_x_)-CT ceramics with an O–T–C multistructure exhibit comparatively poor piezoelectric properties.

### 2.2. (Li, Na, K)(Nb, Sb)-SrZrO_3_ Ceramics with an O–T–PC (or O–T–R) Multistructure

The 0.96(Li_x_Na_0.5−x_K_0.5_)(Nb_0.945_Sb_0.055_)-0.04SrZrO_3_ [(L_x_N_0.5−x_K)NS-SZ] piezoceramics (0.0 ≤ x ≤ 0.05) with 1.0 mol% CuO were densified at 1020 °C and showed a homogeneous perovskite phase with a dense microstructure. Rietveld analysis was carried out on the XRD patterns of the (L_x_N_0.5−x_K)NS-SZ piezoceramics (0.0 ≤ x ≤ 0.05) to determine their crystal structure, as displayed in Figure 3a–c. The ceramic (x = 0.0) exhibited an O–R multistructure consisting of *Amm2* O and *R3m* R structures (Figure 3a). The PC structure observed in the (L_x_N_0.5−x_K)NS-SZ piezoceramics was identified as the *R3m* R structure [77]. The *P4mm* T structure appeared in the ceramic with x = 0.03, which indicates that this ceramic has an O–T–R multistructure (Figure 3b). When x exceeded 0.03, the R (or PC) structure disappeared, and the ceramic (x = 0.05) exhibited an O–T multistructure containing *P4mm* T and *Amm2* O structures (Figure 3c). *T_R–O_* changed to *T_O–T–R_* and *T_O–T_* with an increase in the Li content. Therefore, the crystal structure of the (L_x_N_0.5−x_K)NS-SZ ceramics changed from the O–R multistructure to the O–T–R and O–T multistructures. The diverse electrical characteristics of the (L_x_N_0.5−x_K)NS-SZ piezoceramics (0.0 ≤ x ≤ 0.05) are displayed in Figure 4. The piezoceramic (x = 0.03), which shows an O–T–R multistructure, has a comparatively small *d*_33_ (330 pC/N) and *k_p_* (0.4) [77]. Therefore, the (L_x_N_0.5−x_K)NS-SZ piezoceramics with an O–T–R multistructure showed low piezoelectric properties.

### 2.3. (K, Na, Li)(Nb, Sb)O_3_-(Ca, Sr)ZrO_3_ Ceramics with an O–T–PC (or O–T–R) Multistructure

The crystal structure and piezoelectric properties of the (K_0.5_Na_0.5-z_Li_z_)(Nb_0.92_Sb_0.08_)-(Ca_0.5_ Sr_0.5_)ZrO_3_ [(KN_0.5-z_L_z_)NS-CSZ] ceramics with 0.0 ≤ z ≤ 0.05 densified at 1060 °C have been investigated [82]. Figure 5a–c shows the Rietveld refinement analysis results of the XRD profiles of the (KN_0.5-z_L_z_)NS-CSZ piezoceramics with 0.0 ≤ z ≤ 0.05. The ceramic (z = 0.0) exhibits an O–R multistructure composed of an *Amm2* O structure (30.8%) and an *R3m* R structure (69.2%), as provided in Figure 5a. The *P4mm* T structure appears in the ceramic (z = 0.03), and therefore, this piezoceramic has an O–T–R multistructure (Figure 5b). The R structure disappears in the ceramic with z = 0.05, which indicates that this piezoceramic has an O–T multistructure (Figure 5c). The PC structure formed in the (KN_0.5-z_L_z_)NS-CSZ piezoceramics is understood as the *R3m* R structure [82]. Figure 6a provides the diverse electrical properties of the (KN_0.5-z_L_z_)NS-CSZ ceramics (0.0 ≤ z ≤ 0.06). All ceramics exhibited a large relative density (≥95% of the theoretical density), suggesting that they were well-densified. The piezoceramic (z = 0.03) showed large *ε^T^*_33_*/ε*_0_, *d*_33_, and *k_p_* values of 3800, 560 pC/N, and 0.49, respectively, indicating that the (KN_0.47_L_0.03_)NS-CSZ ceramic with an O–T–R multistructure had excellent piezoelectric properties. Moreover, it provided a high electric field-induced strain of 0.16% at 4 kV/mm (Figure 6b), and therefore, this ceramic is a suitable Pb-free piezoceramic for use in piezoelectric actuators. 

Multilayer ceramics composed of five layers of the (KN_0.47_L_0.03_)NS-CSZ thick film are fabricated (Figure 7a) and utilized to produce a cantilever-style piezoelectric actuator (Figure 7b) [82]. The accelerations are obtained at different frequencies, and the largest acceleration of 42.96 G is detected at 57 Hz (resonance frequency) and ±250 V/mm (Figure 7c). The displacement of the actuator is measured at various electric fields at 57 Hz, as shown in Figure 7d. The displacement increased with an increase in the applied electric field, and a large displacement of 3500 μm was obtained at ±250 V/mm. This displacement is larger than that of the piezoelectric actuators fabricated using PZT- and KNN-modified piezoceramics [82]. Hence, the (KN_0.47_L_0.03_)NS-CSZ piezoceramic can be used in various piezoelectric actuators.

## 3. KNN-Modified Piezoceramics with the R–O–T Multistructure

### 3.1. (1−x)(K, Na)(Nb, Sb)O_3−x_(Bi, Na)HfO_3_ Piezoceramics with the R–O–T Multistructure

Adding the ABO_3_-type substance to the KNNS ceramic increased *T_R–O_* and decreased *T_O–T_* simultaneously; therefore, an R–O–T multistructure could exist [120]. The O structure had 12 spontaneous polarization (*P_S_*) directions along the {110} direction, whereas the R structures had eight possible *P_S_* directions along the <111> direction. The T structure had six possible *Ps* directions along the <100> directions [118], and therefore, the formation of the R–O–T multistructure considerably reduced the energy barrier for polarization rotation in KNN-modified piezoceramics, which resulted in high piezoelectric properties. The (Bi_0.5_Na_0.5_)HfO_3_ (BNH) perovskite phase was added to the (K_0.48_Na_0.52_)(Nb_0.96_ Sb_0.04_)O_3_ (KNNS) ceramic to form an R–O–T multistructure [122]. The (1−x)KNNS-xBNH piezoceramics were formed using a conventional solid-state reaction technique. A two-step sintering process was used to densify the (1−x)KNNS-xBNH piezoceramics, wherein the temperature was raised to 1160–1180 °C, maintained for 1 min, and then cooled down to 1040–1065 °C and kept for 15 h before cooling in the air [122].

The (200) reflections at approximately 45.5° were obtained from the unpoled piezoceramic (x = 0.045) and theoretically fitted with five Lorentz curves, as shown in Figure 8a. Tetragonal (020)_T_ and (200)_T_ peaks, orthorhombic (200)_O_ and (020)_O_ peaks, and rhombohedral (200)_R_ peaks were observed, indicating that the specimen with x = 0.045 has the R–O–T multistructure. Similar results were attained for the (200) reflections of the poled piezoceramics (Figure 8b). For the unpoled ceramic, the intensities of the T reflections were higher than those of the O and R reflections; however, for the poled ceramic, the intensities of the orthorhombic reflections were higher than those of the T and R reflections. Identical results were also obtained from the (1−x)KNNS-xBNH ceramics (0.035 ≤ x ≤ 0.047), which indicate that these piezoceramics have R–O–T multistructures. Table 1 lists the various physical properties of the (1−x)KNNS-xBNH piezoceramics with 0.035 ≤ x ≤ 0.047. The *d*_33_ of the ceramic with x = 0.035 was low, at 350 pC/N, and it enhanced with the increase in x to the largest value of 540 pC/N for the piezoceramic with x = 0.045, and then, it decreased to 475 pC/N for the ceramic (x = 0.047). The ceramic with x = 0.045 had a relatively large *k_p_* value of 0.56, which indicates that the 0.955 KNNS-0.045 BNH ceramic with the R–O–T multistructure showed promising piezoelectric properties (*d*_33_ = 540 pC/N and *k_p_* = 0.56).

### 3.2. (K, Na)(Nb, Sb)O_3_-Bi(Na, K)ZrO_3_-Fe_2_O_3_-AgSbO_3_ Piezoceramics with an R–O–T Multistructure

The existence of the R–O–T multistructure reduces the energy barrier for polarization rotation, thereby enhancing the piezoelectric properties. Furthermore, the presence of polar nanoregions (PNRs) in the KNN-modified ceramics improved their piezoelectric properties because the domain boundary energy decreased with a decrease in domain size, which led to easy domain rotation. Nanodomains are observed in relaxor ceramics [107]. Figure 9a–c shows the schematics of PNRs developed in various relaxor ceramics [107]. In traditional relaxor ceramics such as Pb(Mg_1/3_Nb_2/3_)O_3_ (PMN) and (Pb, La)(Zr, Ti)O_3_, PNRs exist in the nonpolar matrix, as shown in Figure 9a. In relaxor–ferroelectric piezoceramics with a morphotropic phase boundary (MPB) structure that includes PMN-PbTiO_3_ and Pb(Zn_1/3_Nb_2/3_)O_3_-PbTiO_3_ ceramics, relaxor properties can be explained by PNRs in a long-range ordered matrix, as shown in Figure 9b. These ceramics exhibited excellent piezoelectric properties. Furthermore, PNRs can exist in KNN-modified ceramics with a nanoscale R–O–T multistructure, as provided in Figure 9c. Such a slush polar state is expected to have a very low energy barrier and induce easy polarization, thereby resulting in excellent piezoelectric properties [107].

The (0.96−x)(K_0.48_Na_0.52_)(Nb_0.95_Sb_0.05_)O_3_-0.04Bi_0.5_(Na_0.82_K_0.18_)_0.5_ZrO_3_-0.4%Fe_2_O_3−x_AgSbO_3_ (KNNS-BNKZ-Fe-xAs) piezoceramics with 1.2% ≤ x ≤ 3.0% were produced, and their microstructure and piezoelectric properties were investigated. Figure 10a shows the *d*_33_ and piezoelectric strain constant (*d*_33_*^*^*) values of KNNS-BNKZ-Fe-xAs piezoceramics, and the piezoceramics (1.5% ≤ x ≤ 3.0%) also show large *d*_33_*^*^* values (650–750) pm/V. In addition, piezoceramics with 1.5% ≤ x ≤ 2.5% show a very large *d*_33_ of ≥ 600 pC/N [107]. The piezoceramic with 1.6% provides the largest *d*_33_ of 650 ± 20 pC/N, which is the largest *d*_33_ value for the untextured KNN-modified piezoceramics developed up to 2020 [107]. This *d*_33_ value is larger than those of typical soft PZT-type ceramics, as shown in Figure 10b; further, this piezoceramic (x = 1.6%) has a relatively large *T_C_* of ~200 °C compared to the other soft piezoceramics [107]. Figure 10c shows the STEM ABF image along the [110] direction obtained from the KNNS-BNKZ-Fe-xAs (x = 1.6%) ceramic, and it shows the polarization rotation from R to O to T phases, indicating the existence of PNRs in the nanoscale R–O–T multistructure. Further, the large *d*_33_ value of this piezoceramic (x = 1.6%) can be explained by the coexistence of the R–O–T nanophases and PNRs (Figure 10c).

### 3.3. (K, Na)(Nb, Sb)O_3_-MZrO_3_-(Bi, Ag)ZrO_3_ Ceramics (M = Ca, Sr, and Ba) with an R–O–T Multistructure

The structural and piezoelectric properties of 0.96(K_0.5_Na_0.5_)(Nb_0.93_Sb_0.07_)O_3_-(0.04−x)(CaZrO_3_)-x(Bi_0.5_Ag_0.5_)ZrO_3_ [KNNS-(0.04−x)CZ-xBAZ] ceramics were investigated to obtain KNN-modified ceramics with high piezoelectricity [121]. Figure 11a–e shows the XRD reflections at ~66.5° detected by a slow-speed scanning technique and deconvoluted using the Voigt function for the KNNS-(0.04−x)CZ-xBAZ ceramics (0.0 ≤ x ≤ 0.04). The ceramic (x = 0.0) had an R–O multistructure (Figure 11a). The T structure appeared for the ceramic (x = 0.01), which indicates that this ceramic had an R–O–T multistructure (Figure 11b). An identical structure is also observed in ceramics with (0.02 ≤ x ≤ 0.03), as shown in Figure 11c,d. Finally, a ceramic (x = 0.04) has an R–T multistructure (Figure 11e), and therefore, ceramics (0.01 ≤ x ≤ 0.03) have an R–O–T multistructure. Figure 11f provides the Rietveld analysis results of the XRD pattern of the KNNS-0.01CZ-0.03BAZ (x = 0.03) ceramic. This ceramic had an R–O–T multistructure consisting of 28% *R3m* R, 34% *Amm2* O, and 38% *P4mm* T structures, suggesting that the amounts of the R, O, and T structures were similar. The piezoceramics (x = 0.01 and 0.02) have an R–O–T multistructure; however, the amount of the *Amm2* O structure is considerably larger than that of the other structures [121]. Therefore, the ceramic (x = 0.03) has large piezoelectric properties because its structure is close to the ideal R–O–T multistructure, wherein the proportions of the R, O, and T structures are similar.

Figure 12a shows the change of the *ε^T^*_33_*/ε*_0_ values with respect to the temperature of the ceramic (x = 0.0) measured at various frequencies. The *ε^T^*_33_*/ε*_0_ value decreases with increasing frequency; however, the variation of the *T_O–T_* can be negligible, which indicates that this ceramic is a normal ferroelectric material. However, *T_O–T_* increases with an increasing frequency for the ceramic with x = 0.03 (Figure 12b), and therefore, this ceramic exhibits relaxor properties, indicating that this ceramic is expected to contain nanodomains. Figure 12c provides a TEM bright-field image of the ceramic (x = 0.03); nanodomains with an average size of 3 × 20 nm were present in this ceramic. The ceramic with x = 0.03 had an ideal R–O–T multistructure and nanodomains with relaxor properties near the *T_O–T_* corresponding to the ferroelectric-to-ferroelectric transition temperature. Therefore, this ceramic (x = 0.03) was expected to exhibit excellent piezoelectric properties. Figure 13a displays the diverse physical characteristics of the KNNS-(0.04−x)CZ-xBAZ piezoceramics (0.0 ≤ x ≤ 0.04) densified at 1090 °C. The relative density of the ceramic (x = 0.0) is comparatively high (~92.5% of the theoretical density), which does not change significantly with increasing x. The *tan δ* values of all specimens are relatively low, ranging between ~3% and 4%. A comparatively large *ε^T^*_33_*/ε*_0_ of 3752 was detected in the ceramic (x = 0.03). Further, the piezoceramic (x = 0.03) has a very large *d*_33_ of 680 ± 10 pC/N because of the existence of an ideal R–O–T multistructure, wherein the three structures had similar proportions. This *d*_33_ is higher than the maximum *d*_33_ value reported in the literature for KNN-modified piezoelectric ceramics (Figure 13b). The *k_p_*s of the piezoceramics shows a trend similar to that observed for the *d*_33_s of the specimens, and the piezoceramic (x = 0.03) has a large *k_p_* of 0.5. Further, the piezoelectric and structural characteristics of the KNNS-(0.04−x)MZ-xBAZ (M = Sr and Ba) ceramics were investigated [118,120]. The KNNS-0.01SZ-0.03BAZ ceramic exhibited a large *d*_33_ of 650 pC/N and *a k_p_* of 0.51, as shown in Figure 14a. This ceramic had an ideal R–O–T multistructure, in which the three structures had similar proportions (Figure 14b). Moreover, this ceramic had nanodomains, as shown in Figure 14c. Therefore, the large *d*_33_ value of the KNNS-0.01SZ-0.03BAZ ceramic can be explained by the existence of an ideal R–O–T multistructure and nanodomains. The KNNS-0.01BZ-0.03BAZ ceramic show a similar piezoelectricity: *d*_33_ = 640 pC/N, *k_p_* = 0.49, and strain = 0.16%, as shown in Figure 14d. The existence of an ideal R–O–T multistructure with nanodomains is responsible for the high piezoelectricity [120].

## 4. [001]-Textured KNN-Based Lead-Free Piezoceramics

### 4.1. [001]-Textured (K, Na, Li)(Nb, Sb, Ta)O_3_-CaZrO_3_ Ceramics with an O–T Multistructure

The enhancement in the piezoelectric properties of the [001]-textured KNN-modified ceramics was influenced by their crystal structure. Therefore, the piezoelectric properties of [001]-textured KNN-based ceramics with various structures were investigated [81,111,112,113,114,115,116,117,118,119]. The 0.99(K_0.49_Na_0.49_Li_0.02_)(Nb_0.97−x_Sb_0.03_Ta_x_)O_3_-0.01CaZrO_3_ [KNL(N_0.97−x_ST_x_)-CZ] ceramics (0.03 ≤ x ≤ 0.25) were textured along the [001] direction using 3.0% NaNbO_3_ (NN) templates using a two-step sintering process [113]. The ceramic tapes containing NN templates were produced using the tape-casting technique and heated from RT to 1180 °C with a heating rate of 3 °C/min, and then, they were rapidly cooled at the cooling rate of 10 °C/min to 1065–1080 °C and held for 10 h. Figure 15a shows the XRD patterns of the [001]-textured piezoceramics (0.03 ≤ x ≤ 0.25). The shapes of the (002) and (200) reflections at approximately 45.5° changed when x increased, thereby implying a transformation of the crystal structure [113]. The crystal structure of the specimens was determined using the ratio of the (002) and (200) peak intensities (I_(002)_/I_(200)_). The ratio is 2 for the O structure; however, it decreases to 0.5 for the T structure. The specimen has an O–T multistructure when the I_(002)_/I_(200)_ ratio is close to 1.0 [113]. The I_(002)_/I_(200)_ value of the [001]-textured piezoceramic (x = 0.25) was approximately 1.45, and therefore, it was considered to have an O–T multistructure [113]. Although the Lotgering factor (LF) of the ceramic (x = 0.25) was relatively low, ~82.1% (Figure 15a), it was textured along the [001] direction. The *ε^T^*_33_*/ε*_0_ versus temperature curves show that the *T_O–T_* is found at ~31° for the ceramic with x = 0.25 (Figure 15b), which confirms that the [001]-textured KNL(N_0.97−x_ST_x_)-CZ ceramic (x = 0.25) has the O–T multistructure. Figure 15c displays the *d*_33_ and *d*_33_*^*^* values of the untextured and [001]-textured KNL(N_0.97−x_ST_x_)-CZ ceramics (0.03 ≤ x ≤ 0.25); the *d*_33_*^*^* values were calculated at 30 kV/mm. The *d*_33_ and *d*_33_*^*^* values of the [001]-textured piezoceramics were higher than those of the untextured piezoceramics. Further, the [001]-textured ceramic (x = 0.25), which had an O–T multistructure, provided the largest *d*_33_ and *d*_33_*^*^* values of 391 pC/N and 578 pm/V, respectively. The *d*_33_ value of the untextured ceramic (x = 0.25) was ~278 pC/N, indicating that the increase in the *d*_33_ value after [001] texturing was not large because of the presence of a large amount of the T structure [113]. 

### 4.2. [001]-Textured (K, Na)(Nb, Sb)O_3_-MZrO_3_ (M = Sr and Ca) Piezoceramics with an O–PC (or O–R) Multistructure

The [001]-textured 0.96(K_0.5_Na_0.5_)(Nb_1−y_Sb_y_)O_3_-0.04SrZrO_3_ [KN(Nb_1−y_S_y_)-SZ] + xNN piezoceramics (0.0 ≤ x ≤ 0.05 and 0.045 ≤ y ≤ 0.08) were produced via a tape-casting method and sintered at 1060 °C for 6 h. Figure 16a shows a scanning electron microscopy image of the NN seeds utilized to texture the KN(Nb_1−y_S_y_)-SZ ceramics. The dimensions of the NN seeds are ~12 × 12 × 1.0 μm^3^ with a high aspect ratio of 12. The XRD patterns of the KN(Nb_0.945_S_0.055_)-SZ + xNN piezoceramics are provided in Figure 16b. A homogeneous perovskite phase was formed in all samples, and the LFs of the textured ceramics were calculated using XRD patterns, as shown in Figure 16c. The ceramic (x = 0.01) showed a low LF of 81%, which increased with an increase in x to 97% for the specimen with x = 0.03, indicating that 3.0% is the optimum amount of NN templates for texturing the specimens.

Figure 17a shows the XRD reflections at 66.5° detected by slow-speed scanning and deconvoluted using the Voigt function for the ceramic (y = 0.065) textured along the [001] direction using 3.0 mol% NN templates. This ceramic has an O–PC multistructure because it provides a pseudocubic (220)_P_ reflection and orthorhombic (004)_O_, (400)_O_, and (220)_O_ reflections. Rietveld refinement analysis has been conducted on the XRD pattern of the [001]-textured ceramic (y = 0.065) to clearly identify its PC structure (Figure 17b). The PC structure was determined to be an *R3m* R structure; therefore, the [001]-textured KN(Nb_0.935_S_0.065_)-SZ (y = 0.065) ceramic has an O–R (or O–PC) multistructure consisting of the *Amm2* O (51%) and the *R3m* R (49%) structures. Furthermore, the [001]-textured ceramic (y = 0.065) has an ideal O–R (or O–PC) multistructure, wherein the amount of the O structure is nearly the same as that of the R (or PC) structure. Figure 17c shows the various physical properties of the [001]-textured KN(Nb_1−y_S_y_)-SZ ceramics (0.045 ≤ y ≤ 0.08) using 3.0 mol% NN templates. The piezoceramic (x = 0.065), which possesses an O–R (or O–PC) multistructure, revealed a large *d*_33_ value of 620 pC/N and a *k_p_* value of 0.53. The *d*_33_ values of untextured KN(Nb_0.935_S_0.065_)–SZ ceramics (y = 0.065) are yet to be reported. However, the *d*_33_ value of the untextured KN(Nb_0.94_S_0.06_)–SZ ceramic (y = 0.06) has been reported to be 265 pC/N; therefore, it can be assumed that the *d*_33_ value of the untextured KN(Nb_0.935_S_0.065_)–SZ ceramic (y = 0.065) is ~260 pC/N [81]. Hence, the *d*_33_ value of the untextured KN(Nb_0.935_S_0.065_)-SZ (y = 0.065) improved by more than twice the original value after [001]-texturing. Further, the large *d*_33_ value (≥580 pC/N) was preserved up to 110 °C, as provided in Figure 17d. Moreover, the [001]-textured KN(Nb_0.91_S_0.09_)–CaZrO_3_ ceramic also showed an ideal O–R (or O–PC) multistructure composed of the *Amm2* O (59%) and the *R3m* R (41%) structures [114]. This ceramic exhibited a large *d*_33_ value of 610 pC/N; therefore, the KN(Nb_1−y_S_y_)-MZ ceramics (M = Sr and Ca) with an ideal O–PC (or O–R) multistructure exhibited a large *d*_33_ value of 610–620 pC/N.

### 4.3. [001]-Textured (K, Na)(Nb, Sb)O_3_-CaZrO_3_-(Bi, K)HfO_3_ Piezoceramics with an O–R Multistructure

The 0.96(K_0.5_Na_0.5_)(Nb_0.965_Sb_0.035_)O_3_-0.01CaZrO_3_-0.03(Bi_0.5_K_0.5_)HfO_3_ (96KNNS-1CZ-3BKH) piezoceramic was textured using x mol% NN templates (x = 0, 1, 3, 5) [115]. Two-step sintering was used to fabricate the [001]-textured 96KNNS-1CZ-3BKH ceramics. The specimens were heated from RT to 1190 °C at a heating rate of 5 °C/min and rapidly cooled at a cooling rate of 10 °C/min to 1090 °C and held for 10 h [115]. Figure 18a provides the XRD patterns of the [001]-textured 96KNNS-1CZ-3BKH ceramics with various amounts of NN templates. All ceramics show a single perovskite phase and the intensity of the [001] reflection is much stronger than those of the other peaks in the textured ceramics and LF was calculated using the XRD patterns. The values of *d*_33_, *k_p_*, and orientation degree F as functions of the amount of the NN seeds are displayed in Figure 18b. The orientation degree F was the same as the LF and the ceramic textured with 3.0 mol% NN template showed the largest orientation degree F of 98%, thereby suggesting an optimized NN template content of 3 mol% [115]. The [001]-textured 96KNNS-1CZ-3BKH piezoceramic using 3.0 mol% NN templates showed a high *d*_33_ (704 pC/N), *k_p_* (76%), and *T_C_* (242 °C), as shown in Figure 18b. Further, these piezoelectric properties are similar to those of soft PZT-5H piezoceramics and considerably larger than those of the lead-free piezoelectric materials reported in the literature (Figure 18c), thereby indicating that this ceramic can replace PZT-based piezoceramics [115].

Figure 19a shows variations in the reflections at ~17.5° and 36° when an electric field is applied to the [001]-textured 96KNNS-1CZ-3BKH piezoceramic. Reflections at 36° can be fitted using the Lorentz function and analyzed as a mixture of O and R reflections, thereby indicating that the [001]-textured KNNS-CZ-BKH piezoceramic has an O–R multistructure. The Rietveld refinement analysis of the XRD patterns and convergent beam electron diffraction patterns of the [001]-textured 96KNNS-1CZ-3BKH piezoceramic indicate that this ceramic exhibits an O–R multistructure [115]. Therefore, the [001]-textured 96KNNS-1CZ-3BKH piezoceramic with the O–R multistructure is considered to possess a large *d*_33_ because eight *P_S_*s contribute to the polarization along the [001] direction when an electric field is applied to the [001]-textured direction, four *P_S_*s along the <110> direction from the O phase, and four *P_S_*s along the <111> direction from the R phase. Furthermore, when an electric field was applied, a monoclinic (M) reflection appeared because of splitting the (002) peak, as shown in Figure 19a. This intermediate M phase disappeared and transformed into the O phase at a higher electric field of 20 kV/cm. The intermediate M phase serves as a bridge, thereby facilitating the polarization rotation between the rhombohedral [112] and orthorhombic [103] polar axes in the (10-1) plane, as shown in Figure 19b, which results in increased piezoelectricity in the [001]-textured 96KNNS-1CZ-3BKH piezoceramic. Figure 19c provides a TEM bright-field image of the untextured 96KNNS-1CZ-3BKH piezoceramic and the large (micrometer or sub-micrometer scale) strip-like domains exist in this ceramic. A lamellar domain configuration composed of a high density of nanodomains is observed in the [001]-textured 96KNNS-1CZ-3BKH piezoceramic, as shown in Figure 19d. The presence of nanodomains with decreased domain wall energy and increased domain wall mobility contributed to the large piezoelectric properties of the [001]-textured 96KNNS-1CZ-3BKH piezoceramic.

### 4.4. [001]-Textured (K, Na)(Nb, Sb)O_3_-MZrO_3_-(Bi, Ag)ZrO_3_ Piezoceramics (M = Ba, Sr, and Ca) with an R–O–T Multistructure Containing a Large Proportion of the R–O Phase

The [001]-textured KNN-modified ceramics, which have an R–O multistructure, are considered to possess outstanding piezoelectricity because the eight *P_S_*s contribute to polarization along the [001] direction with the application of an electric field along the [001]-textured direction. However, the *d*_33_ values of the KNN-modified piezoceramics with the R–O multistructure are lower than those of the KNN-modified piezoceramics with the R–O–T multistructure [81]. Therefore, the improvement in the piezoelectricity of the [001]-textured KNN-modified ceramics with the R–O multistructure may be limited. By contrast, KNN-modified piezoceramics with an R–O–T multistructure possess large *d*_33_ values [118,120,121,122], and therefore, the [001]-textured KNN-based ceramics with the R–O–T multistructure can exhibit larger *d*_33_ values by controlling the amount of the R–O phase. Textured ceramics are anticipated to possess large *d*_33_ values when produced with an R–O–T multistructure with a large amount of the R–O structure.

The 0.96(Na_0.5_K_0.5_)(Nb_0.93_Sb_0.07_)O_3_-(0.04−x)BaZrO_3−x_(Bi_0.5_Ag_0.5_)ZrO_3_[NKNS-(0.04−x)BZ-xBAZ] ceramics were textured along the [001] direction using 3.0 mol% NaNbO_3_ templates [118]. The XRD patterns of [001]-textured piezoceramics (0.0 ≤ x ≤ 0.04) and the LF of each specimen are provided in Figure 20a. All specimens showed a large LF (>97%), implying that the NKNS-(0.04−x)BZ-xBAZ piezoceramics have been well-textured toward the [001] orientation. Figure 20b presents a HRTEM lattice image of the [001]-textured KNNS-0.02BZ-0.02BAZ ceramic. This piezoceramic has nanodomains (1.5 nm × 10 nm) that contribute to the improvement of the piezoelectric properties of this textured ceramic. Figure 20c provides the EBDS and (001) pole figure images of the untextured NKNS-(0.04−x)BZ-xBAZ (x = 0.02) piezoceramic. The EBSD image provides randomly oriented grains, and the (001) pole figure image showed arbitrarily distributed high-intensity spots, thereby suggesting that this piezoceramic was untextured. EBSD and (001) pole figure images of the KNNS-0.02BZ-0.02BAZ piezoceramic textured along the [001] direction are displayed in Figure 20d. As indicated by the red regions, most grains had a [001] orientation. The black areas represent defects developed in the [001]-textured piezoceramic. Furthermore, a strong spot was observed at the center of the (001) pole figure image, confirming that this piezoceramic was well-textured toward the [001] direction.

A Rietveld refinement analysis was conducted on the XRD patterns of the [001]-textured KNNS-(0.04−x)BZ-xBAZ ceramics (0.0 ≤ x ≤ 0.04), as shown in Figure 21a–e [118]. The ceramic (x = 0.0) had an R–O multistructure with an *R3m* R structure (29%) and an *Amm2* O structure (71%), as shown in Figure 21a. The R–O–T multistructure was formed in the textured ceramics (0.01 ≤ x ≤ 0.03), as shown in Figure 21b–d. The [001]-textured piezoceramic (x = 0.02) possessed an R–O–T multistructure with a 28% R structure, 52% O structure, and 20% T structure (Figure 21c). However, the textured piezoceramic (x = 0.03) had an ideal R–O–T structure, in which the proportion of R, O, and T structures was ~33% (Figure 21d). Finally, an R–T multistructure was developed in the textured piezoceramic with x = 0.04, as shown in Figure 21e.

Figure 22a provides the diverse physical characteristics of the KNNS-(0.04−x)BZ-xBAZ ceramics (0.0 ≤ x ≤ 0.04) textured toward the [001] orientation. The relative densities of the [001]-textured piezoceramics range between 92% and 94% of the theoretical density. The *ε*^T^_33_*/ε*_0_ values of the textured ceramics range between 2100 and 2500, with a comparatively low *tan δ* (0.03–0.045). The *k*_p_ of the textured ceramic (x = 0.0) was ~0.65 (Figure 22a). An improved *k_p_* of 0.68 was obtained from the [001]-textured ceramic (x = 0.01) and it decreased to 0.55 for the piezoceramic with x = 0.04. The [001]-textured ceramic with x = 0.02 retained a high *k*_p_ of 0.64. The [001]-textured ceramic (x = 0.0) possessed a relatively large *d*_33_ value of 630 pC/N, which increased with the increase in x. The textured piezoceramic (x = 0.02) provided a very high *d*_33_ of 805 pC/N, which is the largest *d*_33_ value for the KNN-related lead-free piezoceramics reported to date [118]. Figure 22b shows the *d*_33_ value for the untextured and [001]-textured KNNS-(0.04−x)BZ-xBAZ piezoceramics (0.0 ≤ x ≤ 0.04). The *d*_33_ value of the textured KNNS-0.04BZ piezoceramic with the R–O structure was relatively small (630 pC/N) because the untextured KNNS-0.04BZ ceramic has a small *d*_33_ of 425 pC/N. The textured KNNS-0.02BZ-0.02BAZ ceramic, which had an R–O–T multistructure with a large fraction of R–O phase ((~80%), as shown in Figure 21c), exhibited a high *d*_33_ (805 pC/N) after texturing owing to the large *d*_33_ (506 pC/N) of the untextured piezoceramic (x = 0.02). The increase in *d*_33_ after the [001]-texturing is insignificant for the ceramic with x ≥ 0.03 (Figure 22b), and therefore, an increase in the *d*_33_ is high for the specimens (x ≤ 0.02); however, it is low for the specimens (x ≥ 0.03).

Figure 22c shows changes in the proportion of the R–O phase and the increasing rate of the *d*_33_ value after [001]-texturing for the KNNS-(0.04−x)BZ-xBAZ ceramics. The increase in the *d*_33_ value can be provided by *d*_33_^T^/*d*_33_^UT^, where *d*_33_^T^ and *d*_33_^UT^ indicate the *d*_33_ values of the textured and untextured piezoceramics, respectively. The amount of the R–O phase reduced with the increase in x; however, a large amount of the R–O phase (>80%) existed in the piezoceramics (x ≤ 0.02). The amount of the R–O phase greatly reduced when x was larger than 0.02. The increasing rate of *d*_33_ increased slightly for piezoceramics (x ≤ 0.02), although the amount of the R–O phase reduced. The increasing rate of *d*_33_ decreased considerably when x was larger than 0.02 (Figure 22c) because of the presence of a large amount of the T phase. Therefore, it can be concluded that the largest piezoelectricity is obtained after the [001]-texturing for the piezoelectric ceramics with an R–O–T multistructure containing a large amount of the R–O phase (~80%). The identical results were observed from the [001]-textured NKNS-(0.04−x)MZ-xBAZ (M = Ca and Sr) piezoceramics, which showed large *d*_33_ values of ~760 ± 20 pC/N [116,117]. Figure 22d displays the *d*_33_ values of the KNN-modified piezoelectric ceramics reported in the literature. The untextured KNNS-0.01CZ-0.03BAZ piezoceramic with an ideal R–O–T multistructure exhibited the largest *d*_33_ value of 680 pC/N among the untextured KNN-modified piezoceramics. The [001]-textured KNNS-0.02BZ-0.02BAZ piezoceramic, which had an R–O–T multistructure with a large fraction of the R–O phase, provided the largest *d*_33_ value of 805 pC/N among the [001]-textured KNN-modified piezoceramics reported to date.

## 5. Conclusions

A common method for increasing the piezoelectricity of KNN-modified ceramics is forming a piezoceramic with a multistructure. An O–T–PC multistructure was observed in NK(N_0.95_S_0.05_)-CT, (L_0.03_N_0.47_K)NS-SZ, and (KN_0.497_L_0.03_)NS-CSZ ceramics. Small *d*_33_ values (275–330 pC/N) were observed for the NK(N_0.95_S_0.05_)-0.04CT and the (L_0.03_N_0.47_K)NS-SZ ceramics; however, the (KN_0.497_L_0.03_)NS-CSZ ceramic exhibited a comparatively large *d*_33_ (560 pC/N) and *k_p_* (0.49). The 0.955 KNNS-0.045 BNH ceramic has an R–O–T multistructure and exhibits high piezoelectricity, with *d*_33_ = 540 pC/N and *k_p_* = 0.56. The KNNS-BNKZ-0.016As piezoceramic had a nanoscale R–O–T multistructure. Moreover, it has PNRs that exist in the nanoscale R–O–T multistructure, which results in very high piezoelectric properties (*d*_33_ = 650 ± 20 pC/N). The KNNS-0.01CZ-0.03BAZ piezoceramic exhibited an ideal R–O–T multistructure, wherein the proportions of the R, O, and T structures were similar. Further, this ceramic showed nanodomains and an extremely large *d*_33_ of 680 ± 10 pC/N, which is the largest *d*_33_ value reported in the literature for the untextured KNN-based lead-free piezoceramics. The KNNS-0.01MZ-0.03BAZ ceramics (M = Sr and Ba), which possessed an ideal R–O–T multistructure with nanodomains, provided similar *d*_33_ values.

The [001]-texturing is another method for improving the piezoelectricity of the KNN-modified ceramics, and enhancing piezoelectricity after [001]-texturing depends on the crystal structure of the ceramics. The [001]-textured KNL(N_0.72_ST_0.25_)-CZ ceramic had an O–T multistructure and a relatively small *d*_33_ of 391 pC/N because of the presence of the T structure. The [001]-textured KNNS-MZ (M = Sr and Ca) ceramics exhibited an ideal O–PC multistructure, wherein the amount of the O structure is similar to that of the PC structure. The PC structure was comprehended as an *R* structure, and these ceramics exhibited a comparatively large *d*_33_ of ~620 pC/N. The [001]-textured 96KNNS-1CZ-3BKH piezoceramic had an O–R multistructure with nanodomains, showing a large *d*_33_ of 700 pC/N and *k_p_* of 76%. The [001]-textured KNNS-0.02BZ-0.02BAZ ceramic had an R–O–T multistructure with a large proportion of the R–O phase (~80%), showing nanodomains. This ceramic shows a *d*_33_ value of 805 pC/N, which is the largest *d*_33_ value reported for KNN-based piezoceramics.

## Figures and Tables

**Figure 1 micromachines-15-00325-f001:**
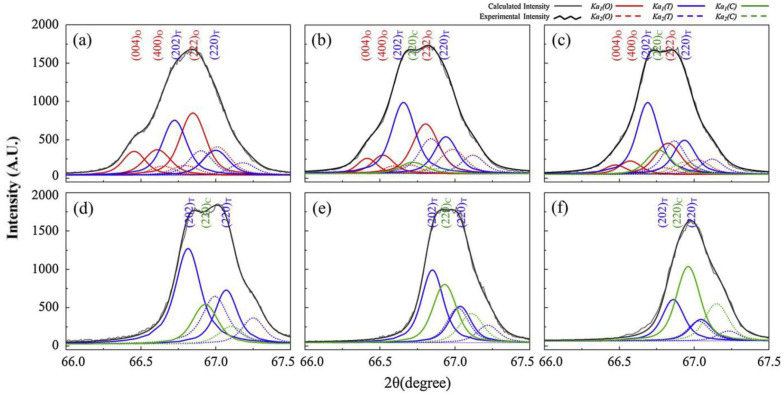
XRD reflections at 67° measured via slow-speed scanning for the NK(N_1−x_S_x_)-0.04CT ceramics sintered at 970 °C for 8 h: (**a**) x = 0.03, (**b**) x = 0.04, (**c**) x = 0.05, (**d**) x = 0.07, (**e**) x = 0.08, and (**f**) x = 0.1 [78].

**Figure 2 micromachines-15-00325-f002:**
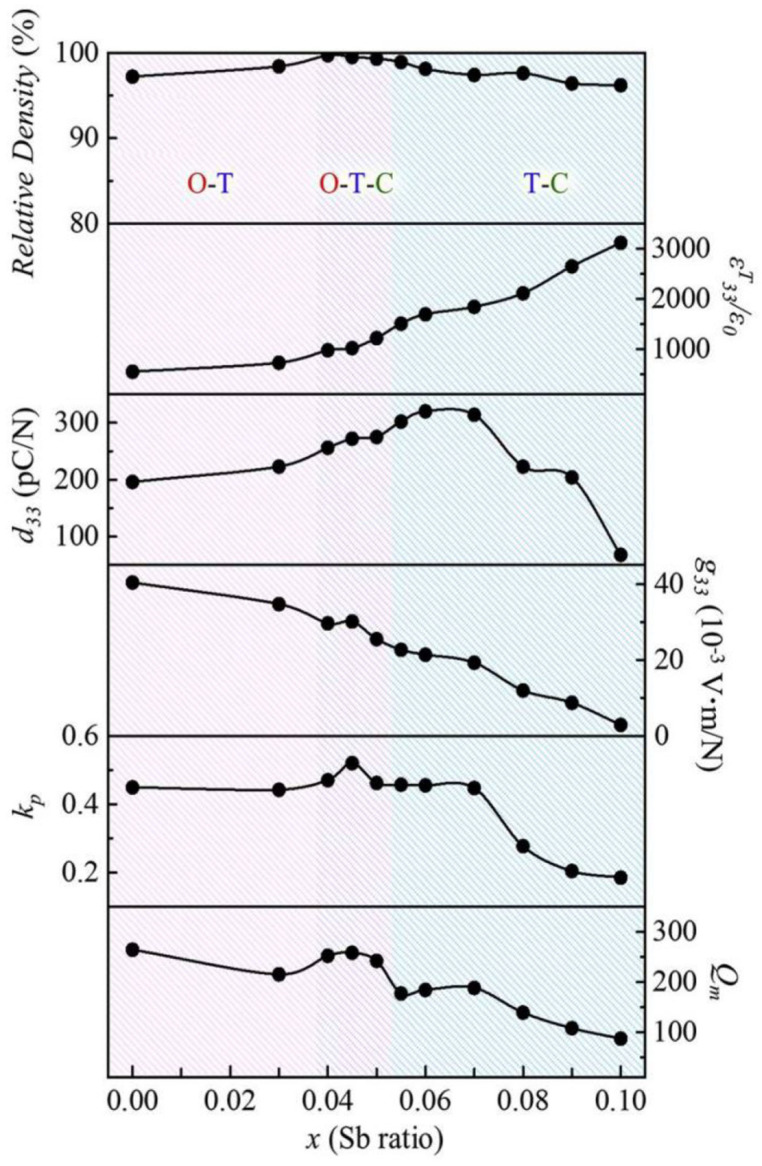
Relative density, *ε^T^*_33_*/ε*_0_, *d*_33_, *g*_33_, *k*_p_, and *Q*_m_ values of the NK(N_1−x_S_x_)-0.04CT ceramics with 0.0 ≤ x ≤ 0.1 [78].

**Figure 3 micromachines-15-00325-f003:**
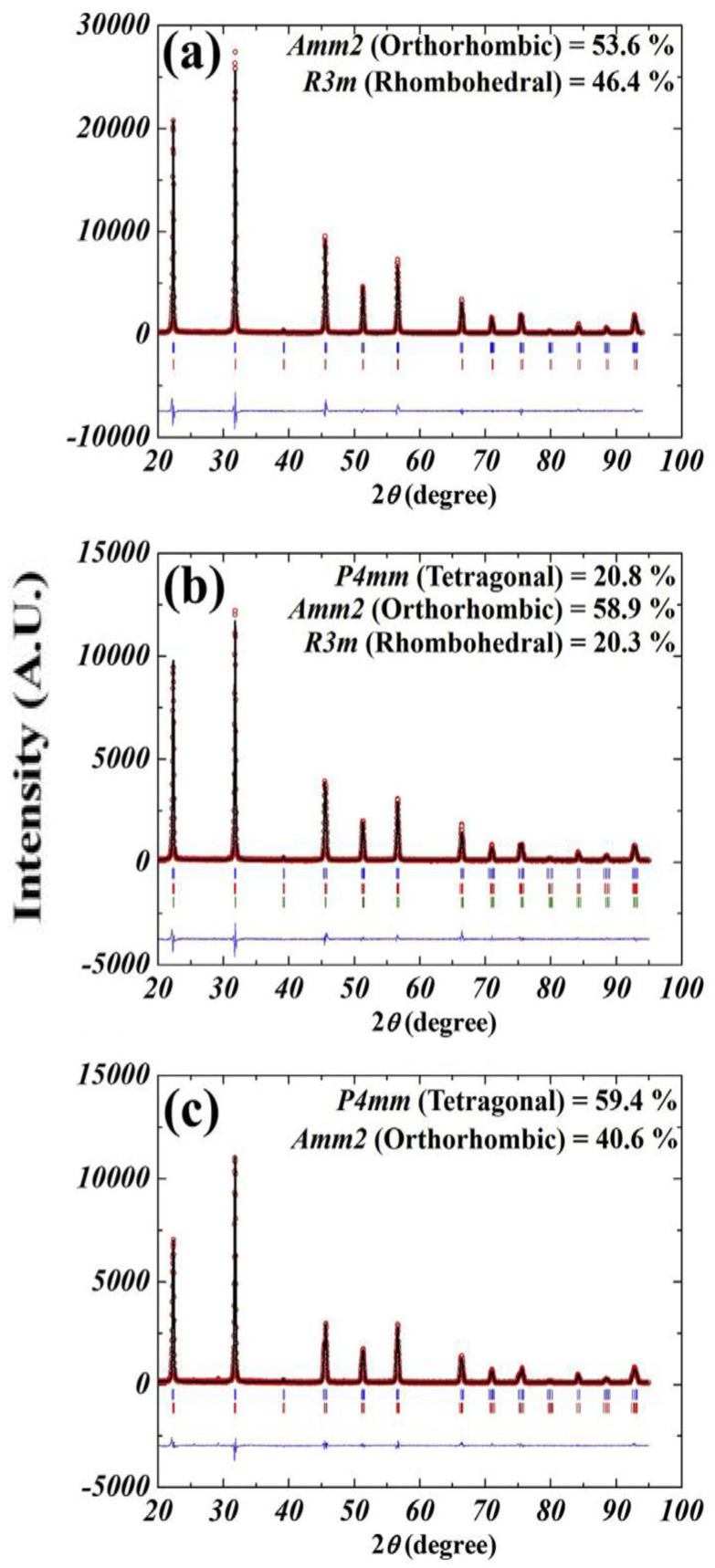
Rietveld refinement XRD profiles of (L_x_N_0.5−x_K_0.5_)(N_0.945_S_0.055_)-0.04SZ ceramics sintered at 1020 °C for 6 h: (**a**) x = 0.0, (**b**) x = 0.03, and (**c**) x = 0.05 [77].

**Figure 4 micromachines-15-00325-f004:**
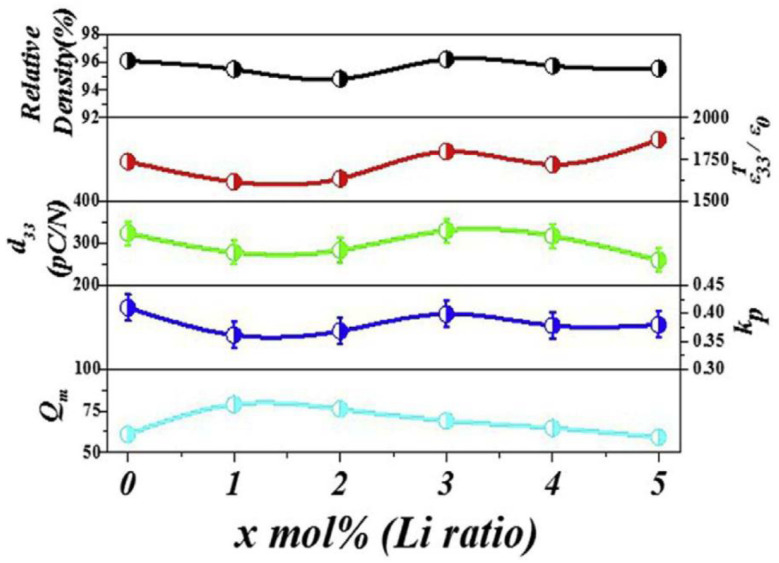
Relative density, *ε^T^*_33_*/ε*_0_, *d*_33_, *k_p_*, and *Q_m_* values of (LxN_0.5−x_K_0.5_)(N_0.945_S_0.055_)-0.04SZ ceramics [77].

**Figure 5 micromachines-15-00325-f005:**
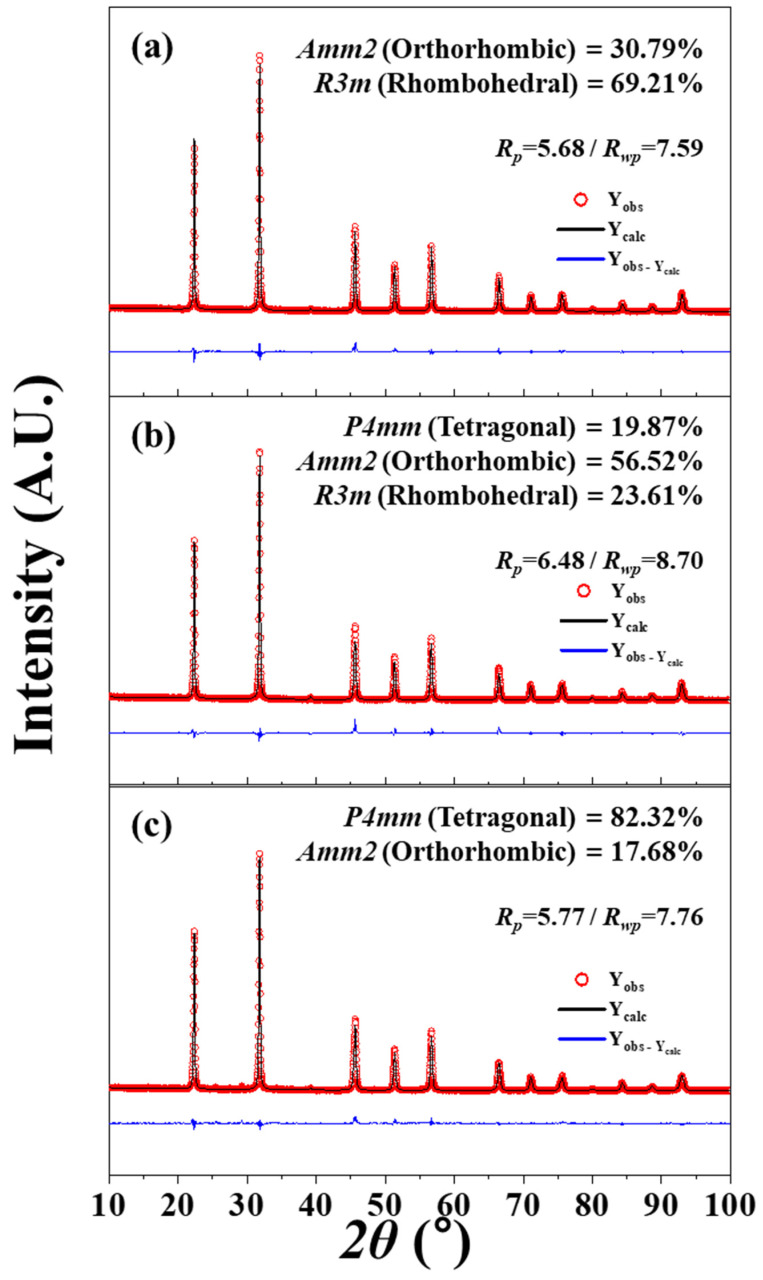
Rietveld refinements of the XRD profiles of (KN_0.5-z_L_z_)NS-CSZ piezoceramics analyzed using various models: (**a**) z = 0.0 (O–R structure), (**b**) z = 0.03 (O–T–R structure), and (**c**) z = 0.05 (O–T structure) [82].

**Figure 6 micromachines-15-00325-f006:**
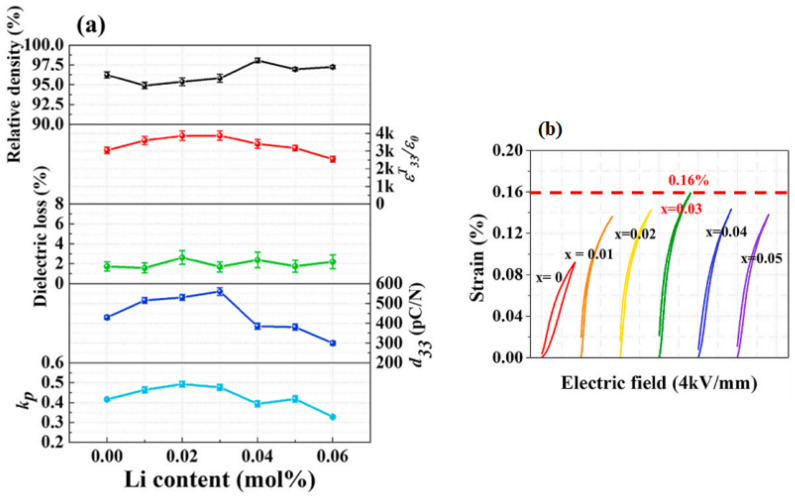
(**a**) Relative density, *ε^T^*_33_*/ε*_0_, dielectric loss, *d*_33_, and *k_p_* and (**b**) unipolar *S–E* curves of (KN_0.5−z_L_z_)NS-CSZ piezoceramics (0.0 ≤ z ≤ 0.06) [82].

**Figure 7 micromachines-15-00325-f007:**
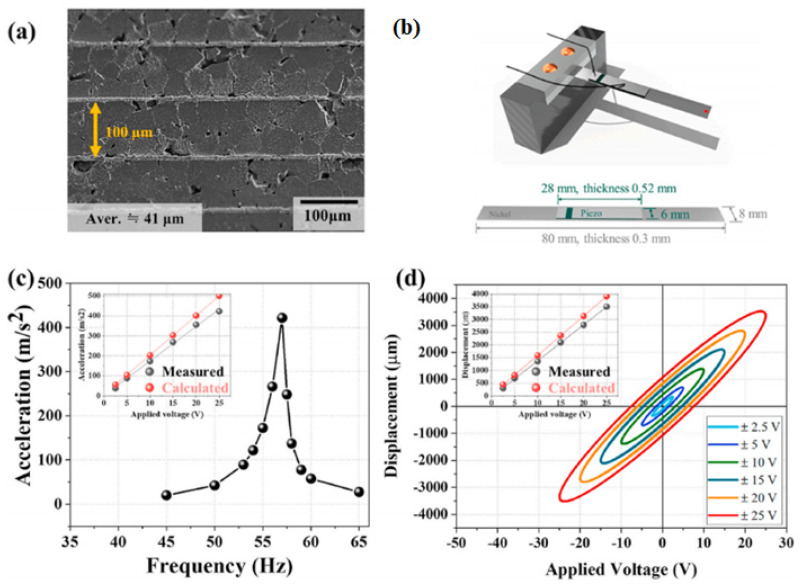
(**a**) SEM image of a (KN_0.47_L_0.03_)NS-CSZ multilayer. (**b**) Schematic of the cantilever-style piezoelectric actuator. (**c**) Change of the acceleration as a function of the frequency measured at 25 V for the (KN_0.47_L_0.03_)NS-CSZ MC actuator. The inset provides accelerations obtained and simulated at different applied voltages. (**d**) Change in displacement with respect to the applied voltage detected at 57 Hz for the (KN_0.47_L_0.03_)NS-CSZ MC actuator. The inset provides the displacements obtained and simulated at diverse applied voltages [82].

**Figure 8 micromachines-15-00325-f008:**
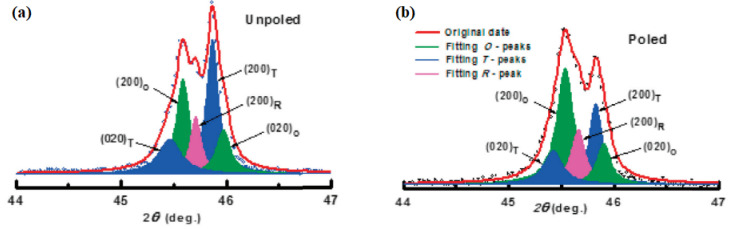
Theoretical fittings of intensity profiles of pseudo-cubic {200} reflection peaks with five Lorentz curves of the (**a**) unpoled and (**b**) poled KNNS-0.045BNH ceramics [122].

**Figure 9 micromachines-15-00325-f009:**
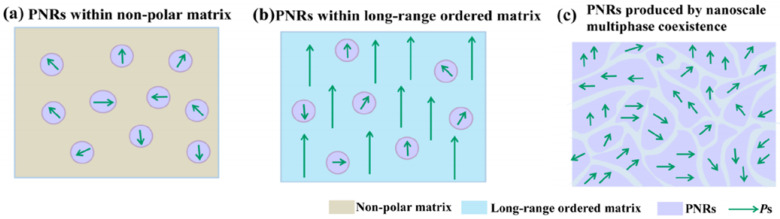
Schematic of PNRs in various relaxor states. (**a**) Conventional relaxor state with PNRs in a nonpolar matrix. (**b**) Relaxor−ferroelectric solid solution with PNRs in a long-range ordered matrix. (**c**) New relaxor state displaying PNRs with multiphase coexistence [107].

**Figure 10 micromachines-15-00325-f010:**
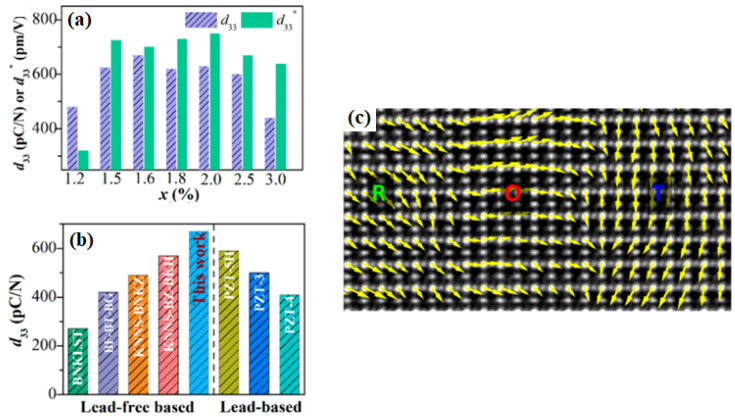
Composition dependence of (**a**) *d*_33_ and *d*_33_^*^ and (**b**) *d*_33_ values of the various lead-free and typical PZT piezoceramics with high T_C_ (>150 °C). (**c**) Atomically resolved contrast-reserved STEM ABF image along the [110] direction showing polarization rotation from R to O to T [107].

**Figure 11 micromachines-15-00325-f011:**
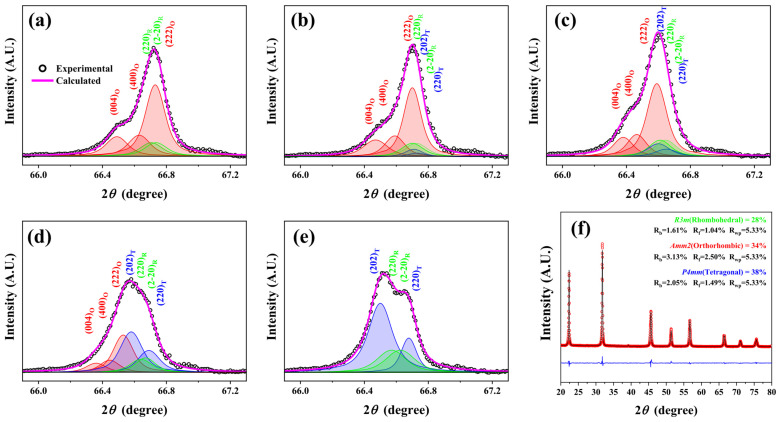
XRD patterns at 2θ~66.5°, obtained by slow-speed scanning and deconvoluted using the Voigt function of the KNNS-(0.04−x)CZ-xBAZ piezoceramics with (**a**) x = 0.0, (**b**) x = 0.01, (**c**) x = 0.02, (**d**) x = 0.03, (**e**) x = 0.04, and (**f**) Rietveld refinement of the XRD pattern of the KNNS-0.01CZ-0.03BAZ piezoceramic based on the R–O–T multistructure [121].

**Figure 12 micromachines-15-00325-f012:**
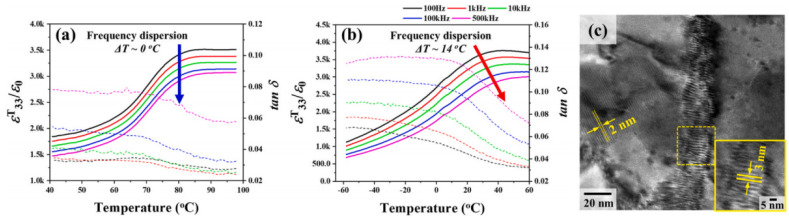
*ε^T^*_33_*/ε*_0_ versus temperature plots measured at various frequencies for the KNNS-(0.04−x)CZ-xBAZ ceramics with (**a**) x = 0.0 and (**b**) x = 0.03. (**c**) TEM bright-field image of the KNNS-0.01CZ-0.03BAZ ceramic [121].

**Figure 13 micromachines-15-00325-f013:**
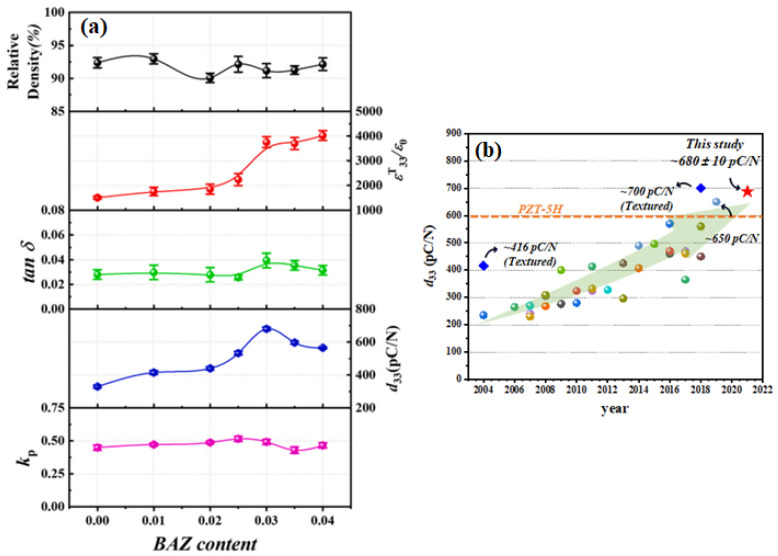
(**a**) Relative density, *ε^T^*_33_*/ε*_0_, *tan δ*, *d*_33_, and *k_p_* of the KNNS-(0.04−x)CZ-xBAZ piezoceramics (0.0 ≤ x ≤ 0.04). (**b**) *d*_33_ values of KNN-modified piezoceramics reported in the literature [121].

**Figure 14 micromachines-15-00325-f014:**
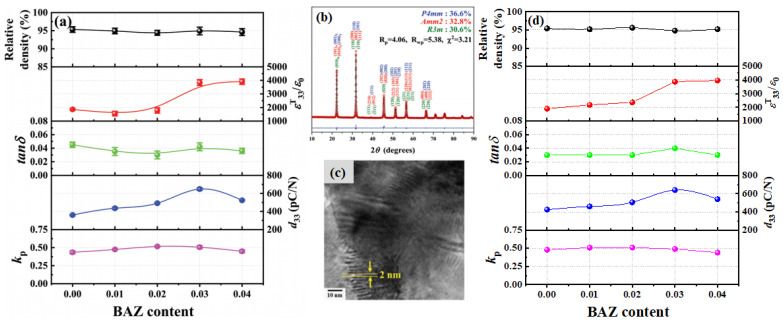
(**a**) Relative densities, *ε^T^*_33_*/ε*_0_, *tan δ*, *d*_33_, and *k_p_* values of the KNNS-(0.04−x)SZ-xBAZ ceramics (0.0 ≤ x ≤ 0.04). (**b**) Rietveld refinement of the XRD profiles for the KNNS-0.01SZ-0.03BAZ piezoceramic using the T–O–R multistructure. (**c**) TEM bright-field image of the KNNS-(0.04−x)SZ-xBAZ piezoceramic (x = 0.03) and (**d**) relative densities, *ε^T^*_33_*/ε*_0_, *tan δ*, *d*_33_, and *k_p_* values of the KNNS-(0.04−x)BZ-xBAZ ceramics (0.0 ≤ x ≤ 0.04) [118,120].

**Figure 15 micromachines-15-00325-f015:**
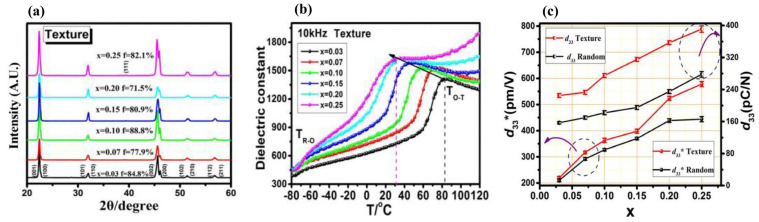
(**a**) XRD patterns, (**b**) *ε^T^*_33_*/ε*_0_ versus temperature curves measured from −80 to 120 °C at 10 kHz, and (**c**) *d*_33_^*^ and *d*_33_ values for the textured and untextured 0.99(KNLN_0.97−x_ST_x_)-0.01CZ ceramics with 0.03 ≤ x ≤ 0.25 [113].

**Figure 16 micromachines-15-00325-f016:**
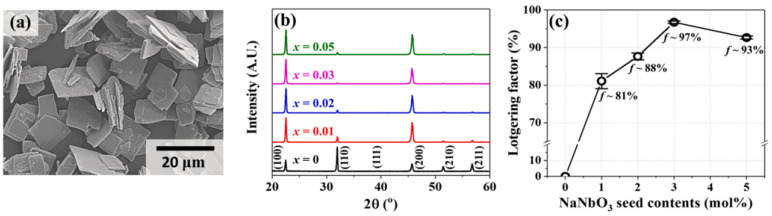
(**a**) SEM image of the NN template. (**b**) XRD patterns and (**c**) Lotgering factors of the KN(N_0.945_S_0.055_)-SZ + *x*NN (0.0 ≤ *x* ≤ 0.05) ceramics sintered at 1060 °C for 6 h [81].

**Figure 17 micromachines-15-00325-f017:**
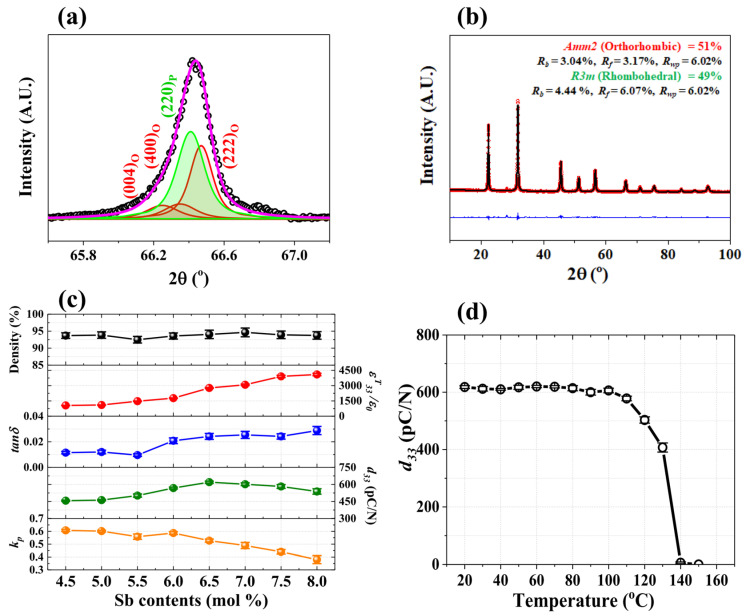
(**a**) XRD reflections at 65.5° measured by slow-speed scanning and deconvoluted using the Voigt function and (**b**) Rietveld refinement of the XRD profile for the KN(N_0.935_S_0.065_)-SZ + 0.03NN ceramic. (**c**) Relative densities, *ε^T^*_33_*/ε*_0_, *tan δ*, *d*_33_, and *k_p_* values of the KN(N_1−y_S_y_)-SZ + 0.03NN ceramics (0.045 ≤ y ≤ 0.08) sintered at 1060 °C for 6 h. (**d**) *d*_33_ value of the sample with *y* = 0.065 obtained at various temperatures [81].

**Figure 18 micromachines-15-00325-f018:**
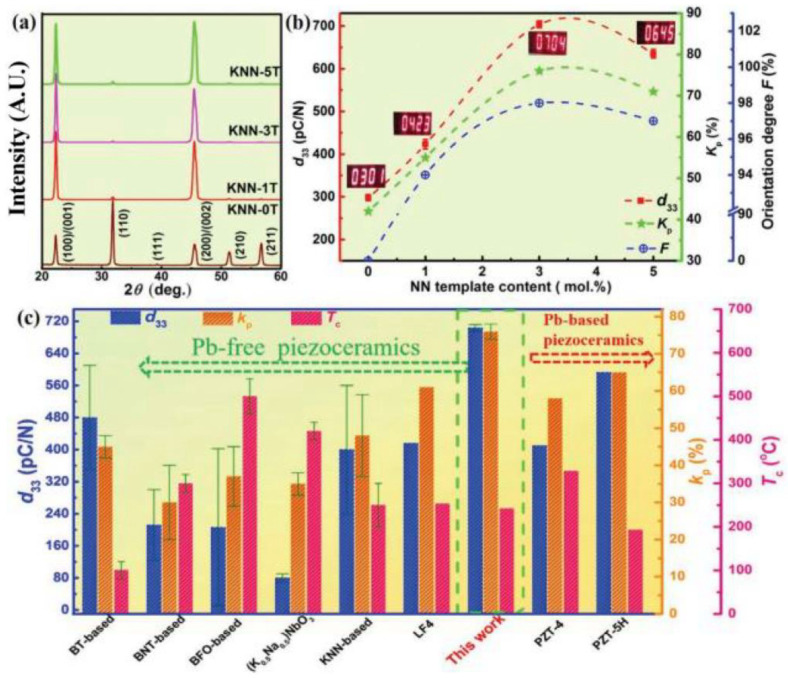
(**a**) XRD patterns of KNNS-1CZ-3BKH + x mol% NN samples (x = 0, 1, 3, 5) detected at RT. (**b**) Orientation degree F, *d*_33_, and *k_p_* values with respect to NN template content. (**c**) Comparison of several important parameters (*d*_33_, *k_p_*, and *T_c_*) in representative lead-based and lead-free piezoceramics [115].

**Figure 19 micromachines-15-00325-f019:**
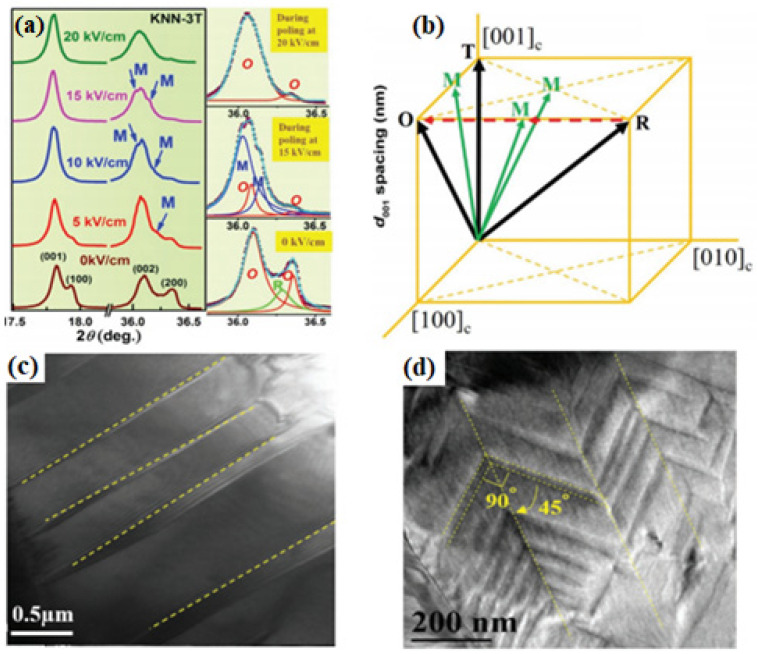
(**a**) In situ synchrotron XRD patterns on (001) and (002) reflections under different electric fields for KNNS-1CZ-3BKH + 3.0 mol% NN ceramic and the (002) peaks with an applied field of 0, 15, and 20 kV cm^−1^ are fitted using a Lorentz function. (**b**) Schematic of the possible polarization rotation path from [112]_R_ to [103]_O_ in (10-1) plane. Bright-field TEM images of domain morphologies for (**c**) untextured and (**d**) [001]-textured KNNS-1CZ-3BKH [115].

**Figure 20 micromachines-15-00325-f020:**
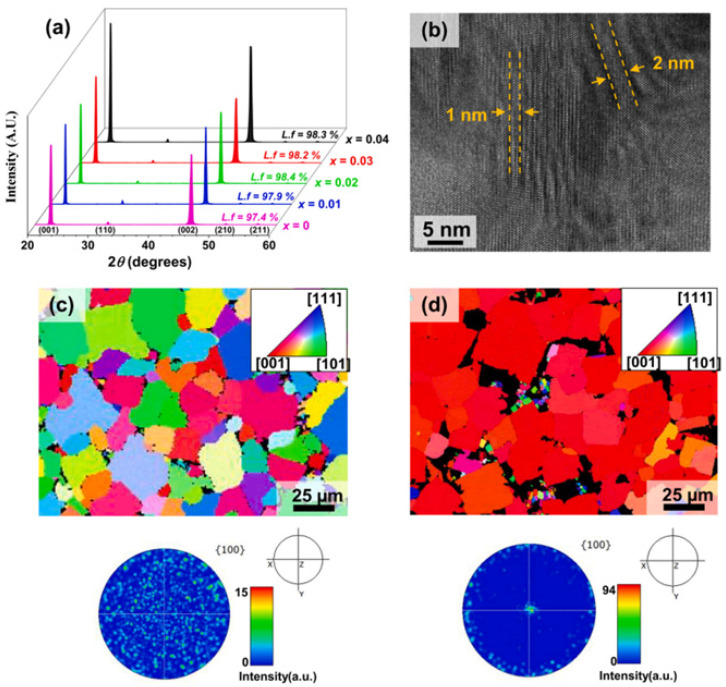
(**a**) XRD patterns of the [001]-textured NKNS-(0.04−x)BZ-xBAZ ceramics (0.0 ≤ x ≤ 0.04) using the 3.0 mol% NN seeds. LFs of the [001]-textured piezoceramics are displayed in this figure. (**b**) HRTEM lattice image of the [001]-textured NKNS-0.02BZ-0.02BAZ ceramic. EBSD and the (001) pole figure images of (**c**) untextured and (**d**) [001]-textured NKNS-0.02BZ-0.02BAZ piezoceramics [118].

**Figure 21 micromachines-15-00325-f021:**
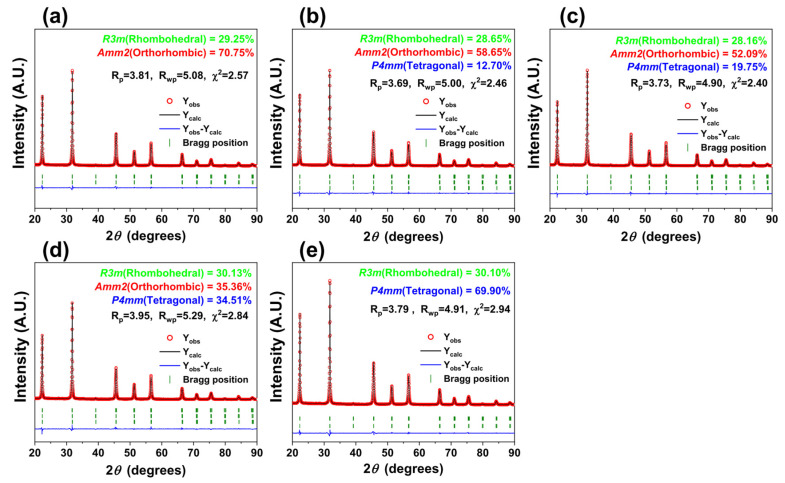
Rietveld refinement of the XRD patterns of [001]-textured KNNS-(0.04−x)BZ-xBAZ piezoceramics: (**a**) x = 0.0, (**b**) x = 0.01, (**c**) x = 0.02, (**d**) x = 0.03, and (**e**) x = 0.04 [118].

**Figure 22 micromachines-15-00325-f022:**
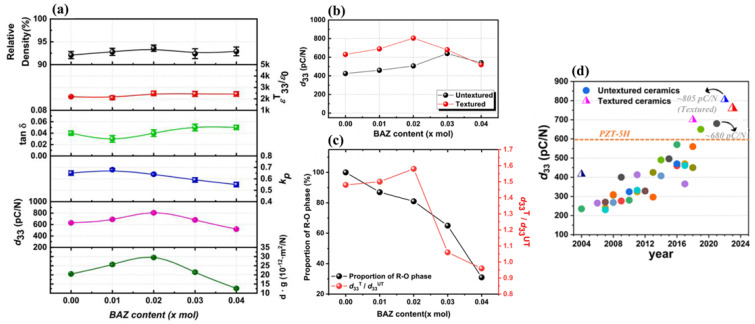
(**a**) Various physical properties of the [001]-textured KNNS-(0.04−x)BZ-xBAZ ceramics (0.0 ≤ x ≤ 0.04). (**b**) *d*_33_ value of the untextured and textured NK(NS)-(0.04−x)BZ-xBAZ piezoceramics (0.0 ≤ x ≤ 0.04). (**c**) Proportion of the R–O phase and the increasing rate of the *d*_33_ value for the [001]-textured KNNS-(0.04−x)BZ-xBAZ piezoceramics (0.0 ≤ x ≤ 0.04). (**d**) *d*_33_ value of the KNN-modified piezoceramics reported in the literature [118].

**Table 1 micromachines-15-00325-t001:** Various physical properties of the (1−x)KNNS-xBNH piezoceramics with 0.035 ≤ x ≤ 0.047 [122].

	*x* = 0.035	*x* = 0.040	*x* = 0.042	*x* = 0.045	*x* = 0.047
*ρ* (g/cm^3^)	4.49	4.56	4.62	4.62	4.61
*d*_33_ (pC/N)	350	450	480	540	475
*k* _p_	0.6	0.58	0.55	0.56	0.53
*ε’*	1550	2150	2680	3350	3340
tan δ	0.03	0.03	0.03	0.03	0.03

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
