# Peer review of "Recent Developments in (K, Na)NbO3-Based Lead-Free Piezoceramics"

_micromachines, 2024, doi:10.3390/mi15030325_

Round 1
Reviewer 1 Report
Comments and Suggestions for Authors
Dear Authors,
The following modifications should be done in the paper before publication. My opinion is that the paper needs a minor revision to be published in the present form. I hope that given comments would be useful for avoiding faults and improving the quality of the paper.
Review:
Title: Recent developments in KNN-based lead-free piezoceramics
· There should be no abbreviations in the title, maybe it should be written differently;
· In Figure 2, what does (a) represent?
· Check the resolution of Figures, the markings in some figures are not very good;
· Check whether too many self-citations are given in the paper;
· Match the marks on the pictures, for examples in the XRD figures, somewhere it says intensity (a.u.), (A.U.), and somewhere (cps). Also, marks (a), (b), are somewhere in the picture somewhere below the Figures.
Reviewer 2 Report
Comments and Suggestions for Authors
In this review, (K0.5Na0.5)NbO3 (KNN) based ceramics with various multiple structures are studied and introduced. Since the piezoelectric properties of [001] textured KNN-based ceramics can be successfully improved, in this review, the [001] textured KNN-based ceramics with different crystal structures are studied and systematically summarized. This work is very meaningful and the following problems should be solved and improved.
(1) The pseudo-cubic structure mentioned in the review is significantly different from the cubic structure, and atomic model diagram can be drawn for specific elaboration.
(2) It is mentioned that multi-phase coexistence and directional grain arrangement can improve the piezoelectric properties of KNN-based ceramics, and relevant literature is listed. The authors need to supplement the mechanism to make the article more readable.
(3) The authors need to supplement the regulation methods of multi-phase coexistence of KNN-based piezoelectric ceramics.
(4) The authors need to supplement the regulatory methods for the texturization of KNN-based piezoelectric ceramics.
Comments on the Quality of English LanguageThe quality of English is relative high, and minor errors need to be revised.
